# Can Botulinum Toxin A Play a Role in Treatment of Chronic Pelvic Pain Syndrome in Female Patients?—Clinical and Animal Evidence

**DOI:** 10.3390/toxins12020110

**Published:** 2020-02-10

**Authors:** Chin-Li Chen, En Meng

**Affiliations:** Division of Urology, Department of Surgery, Tri-Service General Hospital, National Defense Medical Center, 325, Section 2, Cheng-Gung Road, Taipei 114, Taiwan; j0921713355@gmail.com

**Keywords:** botulinum toxin A, chronic pelvic pain syndrome, pelvic pain

## Abstract

Chronic pelvic pain (CPP) is defined as chronic pain and inflammation in the pelvic organs for more than six months. There are wide ranges of clinical presentations, including pelvic pain, painful intercourse, irritable bowel syndrome, and pain during urinating. Chronic pelvic pain syndrome (CPPS) is a subdivision of CPP, and the pain syndrome may be focused within a single organ or more than one pelvic organ. As there is uncertain pathogenesis, no standard treatment is currently available for CPPS. Botulinum toxin A (BoNT-A) is a potent neurotoxin that blocks acetylcholine release to paralyze muscles. Intravesical BoNT-A injection can reduce bladder pain in patients with interstitial cystitis/bladder pain syndrome. BoNT-A injected into the pelvic floor muscles of women has also been reported to improve chronic pain syndrome. Due to the reversible effect of BoNT-A, repeated injection appears to be necessary and effective in reducing symptoms. Adverse effects of BoNT-A may worsen the preexisting conditions, including constipation, stress urinary incontinence, and fecal incontinence. This review summarizes the evidence of BoNT-A treatment for CPPS in animal studies and clinical studies regarding the therapeutic effects of BoNT-A for CPPS in female patients.

## 1. Introduction

Chronic pelvic pain (CPP) is chronic or persistent pain perceived in pelvic structures for more than six months with continuous or recurrent pelvic pain as well as with symptoms suggestive of the lower urinary tract, sexual, bowel, pelvic floor or gynecological dysfunction in men and women [1]. CPP may be divided into situations with well-defined classical pathology (such as infection or cancer) and those without confirmed etiology [1]. According to this classification, the former is considered to be “specific disease-associated pelvic pain,” and the latter is described as “chronic pelvic pain syndrome” (CPPS) [1]. The incidence rate of CPP in women is around 6% to 27% worldwide [2]. CPP in women is reported to affect 18–50-year-old women, mostly [3].

CPPS is a subdivision of CPP and the pain syndrome may be focused within a single organ or more than one pelvic organ, including bladder, urethra, vagina, rectum, anus and whole pelvic musculatures [1]. When the pain is localized to a single organ, some specialists use an end organ term such as Bladder Pain Syndrome (BPS), and use “syndrome” to indicate pain localized in more than one organ site [1]. However, some specialists sub-divide pelvic pain through psychological and functional symptoms rather than anatomy [1].

CPPS of women can exhibit different symptoms, including dyspareunia, dysmenorrhea, dyschezia, and non-menstrual pelvic pain. It may also cause lower urinary tract symptoms, such as frequency, urgency, difficulty urinating, and pain with urination. The most important evaluations for diagnosis of CPPS are medical history and physical examinations. [4]. Vaginal examination is critical to diagnose pelvic floor spasm due to the easily palpable taut muscles. Electromyography, perineometry, vaginal manometry, and digital assessment of pelvic floor muscle (PFM) are also conducted for diagnosis [5]. There are limited laboratory testing and imaging for diagnosis, and it should be considered to be evaluated by laparoscopic or urologic evaluation according to the clinical findings [4].

There is no single definitive etiology or standard management for CPPS. It is supposed that CPPS results from a combination of risk factors, such as neurological, mechanical, and biochemical factors. More than one-half of CPP patients live with interstitial cystitis/painful bladder syndrome (IC/BPS), endometriosis, irritable bowel syndrome, or pelvic adhesions [6,7]. Spasticity of the PFMs, which leads to an increase in muscle tone, has been proposed to play one of the important pathogenic factors of CPPS [8]. PFMs spasm may result from a primary event on the pelvic floor musculatures or secondary to other diseases related to psychological or pathological disorders. It is another significant problem that may decrease quality of life and increase health care costs.

The European Association of Urology (EAU) proposed comprehensive guidelines about the diagnosis and management of CPPS [1]. Management for CPPS remains limited, and the treatment points are usually symptom relief [9]. Secondary disease processes associated with CPPS in women should receive targeted treatment first. The etiologies of CPPS involve multiple mechanisms, so treatments for CPPS need a holistic approach including behavioral, physical, psychological, and sexual components. Modalities of treatments for CPP involve behavioral interventions, physical therapy, medications, surgical interventions, and alternative therapies [4]. The EAU guidelines suggest simple analgesics, such as nonsteroidal anti-inflammatory drugs, to be the first-line therapy of general management. Opioids and neuropathic analgesics, such as tricyclic antidepressants, anticonvulsants, and gabapentin, should be used as further medications if simple analgesics fail. If medications fail to provide symptom relief, nonpharmacologic managements, such as nerve blocks, suprapubic transcutaneous electrical nerve stimulation, sacral neuromodulation, and injection with Botulinum toxin A (BoNT-A) should be considered to help treat CPPS.

Botulinum neurotoxin (BoNT) comprises seven different serotypes (A to G) and more than 40 subtypes [10]. These serotypes have a similar mechanism to inhibit the release of acetylcholine (ACh), but they have different potency of actions. Among different subtypes of BoNT-A, A1 is the only isoform for the therapeutic purpose because of the high potency and long duration of paralysis [11]. The only subtypes of BoNT for clinical purposes are A1 and B1 [12]. However, the paralytic effect of BoNT-B is not as efficient as the function of BoNT-A according to the results of Sloop et al. [13]. The most studied BoNT for clinical treatment is BoNT-A.

BoNT-A injection can reduce spasms and pressure in the PFMs of women with CPPS [14]. Many women with CPPS reported diminution of pelvic pain symptoms after BoNT-A injection. Reduction in pelvic pain can improve quality of life, social activity, working performance, sexual relationships, urinating pain, and mood situations [15]. BoNT-A injected into the pelvic floor musculature of women with CPPS has been reported to improve chronic pelvic pain symptoms and spasms of PFMs [14]. The organ specificity of pelvic pain syndrome in CPPS includes urology, gynecology, gastroenterology, neurology, sexology, and the pelvic floor [1]. This review will focus on the BoNT-A treatment in female CPPS, especially in PFMs pain, IC/BPS, and sexual pain syndrome.

## 2. Mechanisms of BoNT-A for Treating CPPS

Bacterium *Clostridium botulinum* was first isolated in 1895. BoNT-A was first purified from bacterium *Clostridium botulinum* in 1928 and its off-label use started in the 70s [16].

There are three major mechanisms of BoNT-A that function on the muscles, neural system, and inflammation to relieve pain symptoms [17]. BoNT-A plays an important function in the reduction of pain symptoms. It is believed that spasms and tenderness of the PFMs are highly associated with CPPS in women [18,19]. BoNT-A injection has been used to paralyze muscles, and its effect is localized, partial, and reversible. After injecting to the PFMs, BoNT-A can reduce the hypertonic pressure and improve pelvic muscle spasms.

BoNT-A is a selective neurotoxin that acts on neuromusculatures. After binding to terminal receptors on the motor neuron, it can inhibit the release of ACh to cause muscle paralysis. BoNT-A inhibits ACh vesicles releasing to the synaptic cleft by cleaving particular proteins, such as SNAP-25 or VAMP, which are essential for binding with ACh vesicles at the presynaptic membrane. Due to the effect of BoNT-A, there is no release of ACh in the synaptic cleft, and it can paralyze the innervated muscles subsequently [20]. This mechanism has been used to relieve the storage of lower urinary tract symptoms of IC/BPS such as frequency and urgency.

Animal studies reported that BoNT-A could inhibit the delivery of several neurotransmitters, such as calcitonin gene-related peptide (CGRP), glutamate, adenosine triphosphate (ATP), and substance P [21,22,23,24,25].

BoNT-A may block these neurotransmitters from releasing muscular nociceptors, and reduce the symptom of muscle pain in patients with CPPS [26]. BoNT-A could also inhibit the contraction of muscles via alpha and gamma motor neurons and block spasms of pelvic floor musculature, which results in relieving the pelvic pain caused by muscle spasms [26]. Current literature has shown that the use of BoNT-A can reduce the hypertonicity of PFMs to improve pain scores from CPPS patients.

In addition, BoNT-A has the analgesic effect of relieving pain symptoms. From the animal and human studies, increased expression of cell membranes receptors, such as the TRPV1 and P2 × 3, in the nociceptors may up-regulate the symptoms of neuralgia [27,28]. BoNT-A has been reported to reduce the expression of TRPV1 in rats with neuropathic pain [29].

After the injection of BoNT-A, paralysis of muscle occurs after 2–5 days [5]. The functional effects can typically last from three to six months [14]. The clinical efficacy of BoNT-A injection for CPPS in women was durable to 24 weeks [30]. This long-term but reversible effect has made BoNT-A an important therapy for a wide variety of neuromuscular diseases.

After formation of antibodies against BoNT-A, the duration of the BoNT-A effect and the therapeutic extent of the maximal treatment effect are usually reduced after a few BoNT-A applications (partial therapy failure) before complete therapy failure occurs [5].

## 3. Clinical Evidence of BoNT-A for Pelvic Floor Muscle Pain in CPPS Women

BoNT-A was first used for therapeutic purposes in the 1960s when Dr. Alan B. Scott, an ophthalmologist, injected BoNT-A into extraocular muscles of rhesus monkies to treat strabismus. His results were successfully replicated in humans [31]. BoNT-A has been widely studied in several therapeutic applications and different diseases. However, previous studies and clinical trials have not confirmed the effect of BoNT-A for relieving myofascial pain in the neck, shoulders, or trunk [17,32].

Although BoNT-A has been widely used for muscular disorders, its role in treating CPPS has yet to be established. Present literature suggests that injecting BoNT-A into the PFMs may improve symptoms in women with CPPS. Joo et al. reported a 50% success rate of BoNT-A injection in anismus for a long-term period [33]. Brin and Vapnek first reported the injection of BoNT-A to treat vaginismus in 1997 [34].

A systematic review including five studies supports that BoNT-A treatment was beneficial for relieving spasms of PFMs due to CPPS [17,35]. BoNT-A administration could relieve symptoms of dyspareunia, dysmenorrhea, non-menstrual pain and dyschezia. After BoNT-A treatment, it also improved the quality of life and sexual activity. The improvement may because of BoNT-A injection causing localized, partial denervation of the PFMs resulting in muscle weakness and reduction of pressure.

CPP can occur at several muscles in the pelvic region such as pubococcygeus, ischiococcygeus, iliococcygeus, piriformis, transverse perinei muscles, and obturator internus. Preliminary studies have reported that BoNT-A injection could be able to decrease levator ani muscle spasms in women with CPP [14,36,37]. Halder et al. reported 200 U of BoNT-A injections into the trigger points of the perineum in 50 women with CPP [38]. The outcomes showed a significant decrease in pelvic pain scores (3.7 ± 4.0 vs. 6.4±1.8, *p* = 0.005), and fewer trigger points (44% vs. 100%, *p* < 0.001). There was an improvement (20.7%) in patients with the previous placement of incontinence sling versus no improvement (0%) in pain (*p* = 0.003).

The major factor of pelvic pain in women with a spasm of PFMs appears to be ischemia due to vessel compression of the muscles, which leads to the release of bradykinin and sensitization or excitation of nociceptors [5]. BoNT-A injection can improve symptoms of CPP by the antispasticity effect on a spasm of PFMs rather than the result of the antinociceptive activity [17]. Injected BoNT-A into PFMs in women with CPP has been reported to improve the overall quality of life significantly, specifically to improve dyspareunia and female sexual function [30].

Because of the reversible effect and natural metabolism by the body, treatment with repeated BoNT-A injection for CPPS is frequently required. A low dosage of BoNT-A was injected in other muscle groups of the body at three-month intervals, as it was unlikely to produce significant side effects [14].

Injection of 300 U of BoNT-A into PFMs may be safe, especially administered to the regions of hyper-spastic muscles [30]. However, the timing for repeat injection, the optimal dilution, and injected dosage are still challenging to maximize the therapeutic effects of BoNT-A in CPPS of women. It is unclear how significantly the pain will be relieved after BoNT-A injection. The outcome of BoNT-A in relieving CPPS needs more large prospective, randomized, controlled studies.

Table 1 lists important clinical studies of BoNT-A injection for relieving pain symptoms of PFMs.

## 4. Clinical Application of BoNT-A in IC/BPS Women

The etiology and pathophysiology of IC/BPS are still unclear. There are many treatments for relieving IC/BPS including analgesic medicine, changing habits of lifestyle, pentosan polysulfate sodium, cystoscopy with hydrodistention, or instillation of dimethyl sulfoxide in the bladder.

Smith et al. first reported that submucosal BoNT-A injection into 20 to 30 sites of the trigone and bladder floor for a total of 100 to 200 U would result in a 69% improvement of IC/BPS patients, including daytime frequency episodes, nocturia, pain, and bladder capacity [39]. In 2009, Kuo et al. reported a prospective, randomized, and controlled study including 67 patients with IC/BPS who failed conventional treatments, comparing intravesical BoNT-A injection in 100 U or 200 U plus hydrodistention with hydrodistention alone. The IC/BPS symptom score significantly improved in all three groups. However, it only showed a significant reduction of pain visual analogue scale (VAS), an increase of functional bladder capacity (FBC) and cystometric bladder capacity in the BoNT-A group [40].

In 2016, a multicenter, randomized, double-blind, placebo-controlled trial compared hydrodistention plus 100 U of BoNT-A injections at 20 sites with injections of normal saline at 20 sites in 60 patients with refractory IC/BPS. A significantly greater reduction of bladder pain symptoms and increased bladder capacity under cystoscopy were observed in the group with BoNT-A injection, with 63% of overall success rates versus 15% in the normal saline group [41].

Compared to single intravesical BoNT-A injection for IC/BPS patients, there were long-term therapeutic effects in repeated injections, such as pain relief, improved bladder capacity, and better success rates for a long-term period. Kuo et al. reported a repeated intravesical injection of 100 U of BoNT-A every six months for up to four times or until symptoms improved in 81 patients with refractory IC/BPS. It showed significant improvement in pain relief, FBC, and daytime frequency after repeated therapy with different BoNT-A injections. Compared to one single injection, it also reported significantly better success rates for four repeated injections (*p* = 0.0242) and three repeated injections (*p* = 0.050) [42]. In a recent prospective study, 104 patients received an intravesical injection of 100 U of BoNT-A and cystoscopic hydrodistention for refractory IC/BPS. Repeated BoNT-A injections were done every six months for two years. The study showed that 56.7% of patients received four injections of BoNT-A and 34% of patients received another fifth injection because of progressive IC symptoms. It showed a better success rate in those patients who completed repeated BoNT-A injections and significant improvements of O’Leary-Sant IC symptom and problem indexes (ICSI, ICPI, OSS), painful VAS, FBC, frequency, and global response assessment (GRA) during 79 months of follow-up [43].

Liu et al. reported the levels of nerve growth factor (NGF) in the bladder tissue were significantly increased in 19 patients with IC/BPS compared with 12 healthy patients. After injections with 100 U or 200 U of BoNT-A followed by cystoscopic hydrodistention two weeks later, they showed a decrease of the NGF mRNA levels and no significant difference compared with the healthy controls [44]. Intravesical BoNT-A injections could improve chronic bladder inflammation, decrease apoptosis, and decrease the level of bladder vascular endothelial growth factor in patients with IC/BPS [45,46].

The injection sites and the numbers of injections are controversial. It has been reported to inject BoNT-A into the trigone and the posterior bladder wall simultaneously, only the trigone, and sites excluding out the trigone. Pinto et al. reported injecting 100 U of BoNT-A into ten trigonal sites compared to saline injections for refractory IC/BPS patients. They showed significant improvements in bladder pain and quality of life [47]. Jiang et al. injected 100 U of BoNT-A into 20 bladder body sites or 10 trigonal sites in 39 IC/BPS patients. After eight weeks from baseline, they showed no significant difference in changes in urinary frequency, voided volume, post-void residual volume, and bladder capacity. No significant difference in decreasing VAS, symptom improvement, and dysuria were also noted [48].

Kuo et al. investigated the therapeutic predictors, such as ICSI, ICPI, FBC, frequency, and first desire to void for successful treatment of BoNT-A injection in the bladder for refractory IC/BPS patients. Successful treatment was defined as GRA ≥ 2 at six months. The success rate was 45.54% at six months. Multivariate logistic regression showed the only therapeutic predictor for successful management was the baseline ICSI. Patients with an ICSI ≥ 12 may indicate a poorer therapeutic outcome of BoNT-A injections [49]. For refractory IC/BPS, the effectiveness and success of BoNT-A injections has been shown.

To summarize, BoNT-A injection has been widely studied for IC/BPS patients and it has become a promising treatment for refractory IC/BPS patients with the combination with hydrodistention (Table 2), although repeated injection may be needed for long-term therapeutic effects.

## 5. Animal Evidence of BoNT-A for IC/BPS

Although there is currently no definite animal model for CPPS, several chemical-induced cystitis animal models have been used to investigate the pathophysiology and develop a new treatment strategy for this disease. Lucioni et al. explored the effect of BoNT-A on the sensory neurotransmitters in chronic and acute injury models of the rat bladder by intraperitoneal injection of cyclophosphamide (CYP) and incubation of the bladder preparation with hydrochloric acid (HCl). The study found that a greater release of neuropeptides substance P (SP) and CGRP caused by acute injury with HCl and suggested that there is a potential therapeutic effect of BoNT-A in the treatment of neurogenic inflammation of the bladder [50].

Cayan et al. used 41 female Sprague-Dawley rats with intravesical instillation of HCl monthly to induce chemical cystitis and maintain chronic inflammation. These rats injected with 2-3 units of BoNT-A into the bladder detrusor and saline as the control group. Urodynamic studies showed that BoNT-A treatments increase the maximum bladder capacity and bladder compliance compared to the control group at the beginning and end of the study. The histological examinations reported similar counts of mast cells and leukocyte infiltration in these two groups. In this animal model of chemical cystitis, injected BoNT-A into the bladder detrusor led to improvements in vesical function which may be an alternative, minimally invasive treatment compared to other surgical modalities for a chronic inflammatory condition to improve deteriorated bladder function [51].

Smith et al. demonstrated intraperitoneally injected 150 mg/kg CYP to induce chronic cystitis in female Sprague-Dawley rats. It showed an increase of voiding frequency and hyperactivity of bladder. Treatment with CYP or BoNT-A did not affect the release of ATP in resting urothelium. Injection of CYP led to hypoosmotic stimulation and an increase of ATP release in chronic cystitis. After BoNT-A treatment, hypoosmotic shock-induced ATP decreased significantly. Cystometry revealed that CYP injection increased non-voiding bladder contraction. BoNT-A instillation markedly reduced non-voiding contraction frequency that was induced by CYP injection. However, neither CYP nor BoNT-A nor a combination of CYP + BoNT-A had any effect on the contraction frequency of bladder voiding. Furthermore, intravesical instillation of BoNT-A did not affect the release of ATP from the serosal side, implying that its effects were confined to the urothelial side of the bladder preparation [52].

Vemulakonda et al. studied the inhibitory function of BoNT-A on afferent pathways of chronic inflammation in the bladder. Among four groups of female Sprague-Dawley rats, namely group 1: saline-treated, group 2: BoNT-A treated, group 3: CYP treated, group 4: BoNT-A and CYP treated, all animals received intravesical protamine sulfate (1%), followed by intravesical BoNT-A or saline, and subsequently CYP or saline-injected intraperitoneally. Compared to saline controls, the study showed an increase of L6 and S1 c-fos immunoreactive cells after CYP treatment. BoNT-A/CYP treated group presented with a significant decrease of L6 and S1 c-fos immunoreactive cells compared with the CYP group. There was no significant difference in presentation between these two groups of saline and BoNT-A alone. Cystometrogram revealed that the increase of the non-voiding intercontractile interval in the BoNT-A/CYP group was more than 10-fold in CYP group. Conclusively, in a CYP animal model of chronic bladder inflammation, intravesical BoNT-A significantly inhibits the afferent neural response without impairing efferent bladder function [53]. Table 3 concludes the animal studies of applying BoNT-A for IC/BPS treatment.

## 6. Clinical Use of BoNT-A for Sexual Pain Syndrome

Women with sexual pain disorders experience genital pain during sexual intercourse occurring at the periods including before, in the process, or after the sexual activity that involves the clitoris, vulva, vagina, and/or perineum, thus causing difficulty in sexual intercourse and personal distress. Dyspareunia is painful sexual intercourse, in which pain can occur over the external genitalia, inside the vagina or deeper pelvis due to numerous medical, physical, social, or psychological causes. Generally, the prevalence of dyspareunia was reported to affect between 8%–21.1% of women [55]. More sexual pain disorders were reported in female patients with CPP than women without CPP [1]. Morrissey et al. reported a prospective pilot open-label study of 21 women with CPP and refractory high-tone pelvic floor dysfunction (HTPFD) under needle electromyography (EMG) guidance with BoNT-A injections [30]. They prepared 300 U of BoNT-A with nonpreserved saline in a 10-mL syringe and attached it to a 12.5 cm disposable monopolar EMG needle electrode. BoNT-A was injected into the spastic PFM trigger point (30 U), other deeper PFMs including pubococcygeus, iliococcygeus, and coccygeus (30 U each as needed), and obturator internus muscles (up to 60 U into each side). Of these 21 female patients, 66.7% had vulvodynia. After treatment, the dyspareunia VAS score showed significant improvement at weeks 12 (5.6, *p* = 0.011) and 24 (5.4, *p* = 0.004) than baseline (7.8). The Female Sexual Distress Scale (FSDS) showed significant improvement of sexual function at 8 weeks (27.6, *p* = 0.005), 12 weeks (27.9, *p* = 0.006), and 24 weeks (22.6, *p* < 0.001) compared with baseline (34.5). Resting pressures and maximum contraction pressures of the vagina as measured by vaginal manometry significantly decreased during all follow-up examinations (*p* < 0.05).

Vulvodynia is characterized as genital pain without clear etiology that may have resulted from sexual intercourse and causes sexual pain disorder. In the general population, the estimated prevalence of vulvodynia ranges from 10% to 28% in reproductive-aged women [56]. Aberrant increase in the number of nociceptors, which causes peripheral hypersensitivity, leads to intraepithelial neural hyperplasia and strong pain in the vestibule, which may be the cause of vulvodynia [12]. Approximately 7–8% of women have experienced vulvodynia by age 40 that limited sexual intercourse [57]. BoNT-A can inhibit the release of ACh from sympathetic neurons and parasympathetic neurons to relieve vulvodynia and improve dyspareunia. A retrospective study recruited seven women aged 28–61 years with intractable genital pain that was refractory to conventional treatment [58]. Twenty units of BoNT-A was injected into the pain sites including the vestibule, levator ani muscle and the perineal body. If the symptoms had not subsided totally, 40 U of BoNT-A was injected repeatedly every two weeks. After BoNT-A injections, pain decreased or disappeared in all patients. The mean VAS score decreased to 1.4 from 8.3 before the treatment, with no recurrence. The study showed improvement of sexual life without significant pain or discomfort during or after sexual activity.

Hedebo et al. used BoNT-A to treat vulvodynia refractory to conventional treatment for at least six months [59]. The cohort consisted of 79 women and each received 100 U of BoNT-A injections. The results showed significant improvements in dyspareunia (7.82 to 5.82, *p* < 0.01), Negative Interference in Quality of Life (NIQL) (7.88 to 6.19, *p* < 0.01) and the cotton swab test (6.81 to 5.50, *p* < 0.01).

High doses of BoNT-A injection seem to have effectiveness in the treatment of vulvodynia related to sexual pain syndrome. A randomized, double-blind, three-arm, placebo-controlled study from June 2008 to September 2014 included 32 women aged 23–35 years with provoked vestibulodynia [60]. They subcutaneously injected BoNT-A 50 U (arm A), 100 U (arm B) or saline (arm C) into the dorsal vulvar vestibulum and evaluated pain scores after three months. They injected 100 U of BoNT-A for persistently symptomatic women. At the 6-month visit, symptomatic patients received a second injection of BoNT-A 100 U in arm C. The results showed no significant differences in pain between these three groups after three months from the initial injection. However, significant improvements were observed among all three arms using the von Frey filaments test. Exploratory analyses reported that repeat injections with 100 U of BoNT-A over six months had a significant reduction of pain including VAS and von Frey filaments. Fifty-eight percent of patients assessable after repeat BoNT-A injections with 100 U had symptom-free or ≥ 2 points improvement of VAS score.

In 2016 Pelletier et al. evaluated in a prospective cohort study the long-term effectiveness of BoNT-A injection for more than two years in 19 women with provoked vestibulodynia [61]. Fifty units of BoNT-A were diluted in 1.0 mL saline solution followed by injection into bilateral bulbospongiosus muscles for a total dose of 100 U. After 24 months, 37% of participants had no pain. After treatment, they showed significant improvements in the VAS, Dermatology Life Quality Index (DLQI) and Female Sexual Function Index (FSFI) scores at 24 months compared to baseline (*p* < 0.0001). Eighteen women (95%) were able to have sexual intercourse after 24 months.

BoNT-A was successfully used in sexual pain syndrome and appeared to have long-term beneficial effects for sexual activity (Table 4). It is important to continue further research for investigating the novel treatments in the sexual pain syndrome of women.

## 7. Adverse Events of BoNT-A on CPPS

The common adverse events of BoNT-A injection into the bladder for IC/BPS are slow urinary flow rate, decreased detrusor pressure, and dysuria [39,62,63].

The most common adverse event of BoNT-A injection into PFMs for CPPS is dysuria. Increasing flatus has also been reported after BoNT-A injection into bilateral puborectalis and pubococcygeus muscles in women with chronic pelvic floor muscle spasms [14].

Adelowo et al. reported several adverse events, including retention of urine, fecal incontinence, constipation, and rectal pain after BoNT-A injection into PFMs (including coccygeus, iliococcygeus, pubococcygeus, puborectalis, obturator, and pyriformis muscles), which would be resolved spontaneously [36]. This might be because the injection sites were close to the sphincters of the urethra and anus.

Since the urethral sphincter and anal sphincter are adjacent to PFMs, BoNT-A injected into PFMs may result in disruption of urethral and/or anal sphincter mechanisms [30]. The adverse effects after BoNT-A injections reported progression of the following preexisting conditions: constipation (28.6%), stress urinary incontinence (4.8%), fecal incontinence (4.8%), and new-onset stress urinary incontinence (4.8%) [30]. Under electromyography (EMG) guidance, a needle provides more precise delivery of BoNT-A to highly spastic trigger points of the PFMs and helps with the avoidance of neighboring sphincter muscles [30].

Dressler et al. reported atrophy of target muscles after repeated injections of BoNT-A into a hyperactive muscle [64]. However, more serious side effects on systemic organs, such as respiratory failure, heart failure, weakness of muscles, or fatigue have not been reported [17].

Although most of the adverse events resulting from BoNT-A treatment are usually self-repairing, it should be clearly explained to the patient before BoNT-A injection. It is important to discuss with the patient the possibility of mild, transient, and reversible adverse effects on musculatures before BoNT-A injection.

There is still no guidelines about single injections or repeat injections, frequencies of repeat injections, an acceptable interval during repeat injections, injected sites and numbers, and maximum dosage of BoNT-A.

## 8. Conclusions

It is challenging to treat female CPPS patients because of the uncertainty of this disorder. There is currently no definite animal model for CPPS, although the chemical-induced cystitis rat model has been used to investigate the treatment strategy for this disease. Combination management, including physical therapy, biofeedback, behavioral modifications, and medicines may improve CPPS in women. Current literature suggests BoNT-A injection provides promising results in relieving symptoms of pelvic floor pain and muscle spasms in female patients. Intravesical injection of BoNT-A plus hydrodistension also helps to improve symptoms in refractory IC/BPS patients. BoNT-A injection appeared to have long-term beneficial effects for sexual activity in patients with sexual pain syndrome. It is safe for BoNT-A injection with limited adverse effects. However, more double-blind, randomized, controlled clinical studies and well-designed animal studies are needed to support the beneficial efficacy of BoNT-A injection in female patients with CPPS.

## Figures and Tables

**Table 1 toxins-12-00110-t001:** Study of BoNT-A for pelvic floor muscle pain in CPPS women.

Study	Javis [14]	Adelowo [36]	Nesbitt-Hawes [37]	Halder [38]	Morrissey [30]
Numbers	12	29	37 (single injection: 26; multiple injection: 11)	50	21
Age	31.1 (18–55)	55 (38–62)	Single injection: 30 Multiple injection: 31 (21–52 years)	44.5	35.1 (22–50)
Study Model	Prospective cohort study	Retrospective cohort study	Prospective cohort study	Retrospective case series	Prospective pilot open-label study
Follow-Up	12 weeks	Visit 1: <6 weeks post-injection Visit 2: ≥6 weeks post injection	26 weeks	6 weeks (2–192 weeks)	6 months
Criteria	Objective hypertonicity of PFM and 2-year history of CPP at least	Refractory myofascial pelvic pain	Objective overactivity of PFM and a two-year history of pelvic pain	CPP, trigger points of pelvic floor on examinations, and failure (with subsequent discontinuation) of one treatment modality at least including outpatient physical treatment and/or oral analgesics	CPP and HTPFD who have failed conventional therapy
Dose of BoNT-A	40 U	100–300 U	100 U	−	Up to 300 U
Injection Sites	Bilateral puborectalis and pubococcygeus muscles	PFMs (coccygeus, iliococcygeus, pubococcygeus, puborectalis, obturator, and pyriformis muscles)	Puborectalis and pubococcygeous muscles	Multiple areas of the perineum	Spastic PFM trigger points and deeper PFMs (pubococcygeus, iliococcygeus, coccygeus, and obturator internus muscles)
Outcomes	Median VAS scores presented improvements on dyspareunia (80 vs. 28, *p =* 0.01) and dysmenorrhea (67 vs. 28, *p* = 0.03). PFMs manometry showed a 37% reduction in resting pressure at week 4 and a 25% reduction maintained at week 12 (*p* < 0.0001). It showed significant improvements of sexual activity scores, with a reduction in discomfort (4.8 vs. 2.2, *p* = 0.02) and improvement in habit (0.2 vs. 1.9, *p* = 0.03).	79.3% improvement in pain. 51.7% female patients elected to have a second BoNT-A injection. The median time of the first injection to the second injection was 4.0 months (3.0–7.0 months).	26 (70%) women had one injection of BoNT-A and 11 (30%) had 2 or more injections. The median number of repeat injections was 3. The second injection was performed at the earliest at 26 weeks after the first, with subsequent injections having a median time to re-injection of 33.4 weeks (range 9.4–122.7 weeks). Single and repeated injections both significantly reduced dyspareunia by VAS scores (54 to 30, and 51 to 23, *p* = 0.001), non-menstrual pelvic pain VAS (37 to 25, *p* = 0.04), as well as vaginal pressures (40 vs. 34 cm H_2_O (*p* = 0.02).	Posttreatment, patients had lower average pelvic pain scores (6.4 to 3.7, *p* = 0.005), and fewer trigger pints (44% vs. 100%, *p* < 0.001)	61.9% improvement on GRA at 4 weeks. 80.9% improvement on GRA at 8, 12, and 24 weeks. Dyspareunia VAS significantly improved at weeks 12 (5.6, *p* = 0.011) and 24 (5.4, *p* = 0.004). Sexual dysfunction as measured by the FSDS significantly improved at 8 weeks (27.6, *p* = 0.005), 12 weeks (27.9, *p* = 0.006), and 24 weeks (22.6, *p* < 0.001) compared with baseline (34.5). Vaginal manometry demonstrated a significant decrease in resting pressures and in maximum contraction pressures at all follow-up visits (*p* < 0.05).

CPP: chronic pelvic pain. PFM: pelvic floor muscle. HTPFD: high-tone pelvic floor dysfunction GRA: global response assessment. FSDS: Female Sexual Distress Scale.

**Table 2 toxins-12-00110-t002:** Study of BoNT-A for IC/BPS women.

Reference	Study Design	Diagnosis	Numbers	Age	Follow-Up	BoNT-A Dose	Assessment	Outcomes
Kuo HC [41]	Multicenter, randomized, double-blind, placebo-controlled trial	IC/BPS refractory to conventional treatment	60 (52 women, 8 men)	50.8	8 weeks	100U (cystoscopic hydrodistention plus intravesical injections of 100 U BoNT-A)	Pain VAS, 3-day voiding diary, ICSI, ICPI, VUDS GRA	Δ pain VAS of BoNT-A group vs. control group: −2.6 vs. −0.9 (*p* = 0.021). 8 weeks after BoNT-A injection, ICSI, ICPI, OSS, GRA, and FBC all showed significant improvement in both groups. At 3 months, GRA in BoNT-A group vs. saline group: 62% vs. 15% (*p* = 0.028).
Kuo HC [42]	Prospective interventional study	IC/BPS refractory to conventional treatment	81 (71 women, 10 men)	Women: 48; Men 48.2	24 months	100 U (injected into bladder walls at posterior and lateral sites) followed by cystoscopic hydrondistention and repeated injections every 6 months up to 4 times	Pain VAS, 3-day voiding diary, ICSI, ICPI, VUDS, GRA	It showed significant improvement in ICSI, ICPI, VAS, FBC, and daytime frequency after repeated treatment of BoNT-A with different injections. It showed better success rates in patients with 3 (*p* = 0.050) and 4 (*p* = 0.0242) repeated injections of BoNT-A, compared to those with a single injection. Dysuria after each injection: 30%. UTI after each injection: 4.9−19%.
Lee CL [43]	Prospective study	Refractory IC/BPS	104 (88 women, 16 men)	Women: 48.5; Men: 46.6	79 months	100 U (delivered at 20 suburothelial locations at posterior and lateral bladder walls) followed by cystoscopic hydrodistention and repeated injections every 6 months up to 4 times or until symptoms resolved	Pain VAS, 3-day voiding diary, ICSI, ICPI, VUDS, GRA	At 6 months after one single injection, improvement of symptoms include overall OSS (23.7 ± 6.1 vs. 16.6 ± 8.9), pain VAS (5.2 ± 2.4 vs. 3.5 ± 2.5), FBC (129.1 ± 75.0 vs. 177.7 ± 85.0), daytime frequency episodes (15.3 ± 7.7 vs. 11.3 ± 6.3), and all of the *p* value <0.0001. Improvement of GRA (1.31 ± 0.97, *p* < 0.0001). After fourth injection of BoNT-A, OSS (24.6 ± 6.1 vs. 15.2 ± 8.9), VAS (5.4 ± 2.2 vs. 2.9 ± 2.3), FBC (133.5 ± 74.0 vs. 226.9 ± 108.8), and daytime frequency (15.2 ± 7.1 vs. 10.3 ± 5.3) all showed improvement with *p* value < 0.0001. After each injection of BoNT-A, the most frequently reported symptom is dysuria (32.7% to 41.7%). After each injection of BoNT-A, 5.9% to 13.9% of patients. occurred urinary tract infection.
Liu HT [44]	Prospective study	Refractory IC	19 (14 women, 5 men)	Women: 37; Men: 41	3 months	100 U (14 patients) or 200 U (5 patients) followed by cystoscopic hydrodistension 2 weeks later	Pain VAS, 3-day voiding diary, VUDS	The FBC of the overall patients increased by 1.4 times the baseline value. After BoNT-A treatment, the pain VAS score decreased from 5.16 ± 2.09 to 2.53 ± 1.43 (*p* < 0.0001), and the daily frequency episodes decreased from 12.6 ± 4.3 to 8.8 ± 2.5 (*p* = 0.001). NGF mRNA levels at baseline: IC patients vs. control=0.65 ± 0.33 vs. 0.42 ± 0.25, p = 0.046). At 2 weeks after BoNT-A injections, the levels of NGF mRNA had decreased to 0.47 ± 0.23 (*p* = 0.002). The overall success rate: 74%.
Shie JH [45]	Prospective study	IC/BPS and glomerulations after cystoscopic hydrodistention	23 women (11 received three repeated injections every 6 months)	46.6	18 months	100 U (40 suburothelial injections at the lateral and posterior bladder walls) followed by cystoscopic hydrodistention	Pain VAS, 3-day voiding diary, OSS, UDS, GRA	After single BoNT-A treatment, it showed improvements in clinical symptoms, pain VAS, and daytime frequency. After single injection of BoNT-A, tryptase decreased significantly. 11 patients who received three repeated injections of BoNT-A showed significantly lower pain VAS (mean: 5.8 vs. 3.03, *p* = 0), glomerulation degree (mean: 1.8 vs. 1.2, *p* = 0.026) and GRA (mean: 0.3 vs. 1.2, *p* = 0). SNAP-25 decreased after repeated injections with BoNT-A.
Peng CH [46]	Prospective study	Refractory IC/BPS	21 (20 women, 1 man)	44.8	24 weeks	100 U (20 suburothelial injection at posterior and lateral bladder walls) with cystoscopic hydrodistention and repeated every 6 months for 4 times	Pain VAS, 3-day voiding diary, OSS, UDS, GRA	After BoNT-A treatment, it showed significantly decreased in OSS (15.1 ± 8.65 vs. 21.1 ± 7.92, *p* = 0.009) and VAS (3.0 ± 2.92 vs. 5.14 ± 2.46, *p* =0.003). Decreased VEGF level: after BoNT-A treatment vs. baseline = 0.83 ± 0.28 vs. 1.0; *p* = 0.016. After BoNT-A treatment, apoptotic cell count decreased from 1.76 ± 1.69 to 0.86 ± 1.00 (*p* = 0.026) and mast cell activity decreased from 5.82 ± 4.97 to 1.81 ± 2.29 (*p* = 0.009).
Pinto RA [47]	Single center, randomized, double-blind, placebo controlled, phase 2 study	IC/BPS	19 women	45.8	12 weeks	100U (10 trigonal sites)	Pain VAS, 3-day voiding diary, OSS, QoL score	At week 12 BoNT-A treatment, it showed significantly reduced pain compared with saline (−3.8 ± 2.5 vs. −1.6 ± 2.1, p <0.05). The mean change in OSS from baseline to week 12: BoNT-A group vs. saline group = −9 ± 4.7 vs. −7.1 ± 4.6, *p* <0.05). Reductions in voiding frequency were observed at BoNT-A group.
Jiang YH [48]	Single center, randomized, double-blind study	Refractory IC/BPS for at least 6 months	39 women (bladder body, n = 20; trigone, n = 19)	53.9 (bladder body group), 55.1 (trigone group)	12 weeks	100U (comparative group: 20 bladder body sites at the posterior and lateral walls; treatment group: 10 trigonal sites) followed by cystoscopic hydrodistention	Pain VAS, 3-day voiding diary, OSS, VUDS, GRA	After BoNT-A injections, thirteen (65.0%) patients in bladder body group and 10 (52.6%) patients in trigone group had improvement of VAS more than 2 points (*p* = 0.43). After BoNT-A treatment, nine (45%) patients in bladder body group and 10 (52.6%) patients in trigone group had GRA ≥ 2 (*p* = 0.63) and dysuria (*p* = 0.52).
Kuo YC [49]	Prospective study	Refractory IC/BPS	101 (88 women, 13 men)	48.45 (Women: 48.81; Men: 46.0)	6 months	100U (20 suburothelial injection at posterior and lateral walls) immediately followed by cystoscopic hydrodistention	Pain VAS, 3-day voiding diary, OSS, ICSI, ICPI, VUDS, GRA	Significant improvements observed in OSS, ICSI, ICPI, pain VAS, FBC, daytime frequency, nocturia, GRA at 3 months after BoNT injections, and these improvements could exist at 6 months. Overall successful rate at 6 months: 45.54% (women: 46.59%; men: 38.46%). Baseline ICSI score was the only significant predictor for a therapeutic outcome (cutoff value of ICSI to predict treatment failure: ICSI ≥ 12).

GRA: global response assessment. VAS: visual analog scale. OSS: O’Leary-Sant symptom score. ICSI: O’Leary-Sant symptom indexes. ICPI: O’Leary-Sant problem indexes. UDS: Urodynamic study. VUDS: videourodynamic study. FBC: functional bladder capacity. NGF: nerve growth factor. SNAP-25: 25-kD synaptosomal-associated protein. VEGF: vascular endothelial growth factor. QoL: quality of life.

**Table 3 toxins-12-00110-t003:** Study of BoNT-A treatment for IC/BPS in the animal.

Reference	Animal Numbers	Models	Dose of BoNT-A	Outcomes
Lucioni [50]	18 male Sprague-Dawley rats (300–350 g)	Intraperitoneal injection with CYP or saline for 10-day	Harvested bladders were incubated in 10 U BoNT-A for 1 h	Neuropeptides SP in saline group vs. CYP group: 1060 vs. 605 pg/g (*p* < 0.005) SP in CYP group: before BTX vs. after BTX: 1060 vs. 709 pg/g (*p* < 0.05)
Smith [52]	21 female Sprague– Dawley rats (200–250 g)	Intravesical instillation and intraperitoneal injection into four groups (n = 5–6 per group): (1) Control (intravesical saline/intraperitoneal saline) (2) BoNT-A (intravesical BoNT-A/intraperitoneal saline) (3) CYP (intravesical saline/intraperitoneal CYP) (4) CYP + BoNT-A (intravesical BoNT-A/intraperitoneal CYP)	Bladder was instilled with 1 mL of 20 U BoNT-A for 30 min	BoNT-A instillation markedly reduced bladder hyperactivity induced by CYP by reducing non-voiding contraction frequency by 91%.
Cayan [51]	41 female Sprague-Dawley rats (200–300 g)	Intravesical instillation of HCl (0.2 mL of 0.4 N HCl) induced chemical cystitis	2–3 U (0.2–0.3 mL) BoNT-A was injected into the detrusor at the 3, 6, 9 and 12 o’clock positions (10–12 sites)	Increases in the maximum bladder capacity and compliance were significantly higher in the BoNT-A group compared to the control group (*p* = 0.000 and *p* = 0.025).
Vemulakonda [53]	24 female Sprague-Dawley rats (200–250 g)	Intravesical instillation and intraperitoneal injection into four groups: (1) Saline (intravesical saline/intraperitoneal saline) (2) BoNT-A (intravesical BoNT-A/intraperitoneal saline) (3) CYP (intravesical saline/intraperitoneal CYP) (4) CYP/BoNT-A (intravesical BoNT-A/intraperitoneal CYP)	Bladder was instilled with 20 U BoNT-A for 30 min	After CYP treated, expression of c-fos increased significantly in L6 and S1 (78% and 107%) compared to saline control (*p* < 0.001). Compared to the CYP group, it showed a significant decrease of c-fos expression in L6 and S1 (50% and 52%) in the BoNT-A/CYP group (*p* < 0.001). Compared to CYP group, the increase of nonvoiding intercontractile interval was more than 10-fold in BoNT-A/CYP group (*p* < 0.01).

Cystitis of rats was induced by chronic CYP model that reported by Vizzard [54]. Intraperitoneal injection with CYP (150 mg/kg) was administered every third day to a total of three doses to achieve chronic inflammation. CYP: cyclophosphamide. HCl: hydrochloric acid. SP: substance P.

**Table 4 toxins-12-00110-t004:** Studies of BoNT-A injection for sexual pain syndrome in women.

Paper	Study Model	Patients	Treatment	Injection Sites	Result Measures	Duration	Outcomes
Yoon [58]	Retrospective study	7	Dilution: 20 U of BoNT-A diluted in isotonic saline. Dose: 20 to 40 U of BoNT-A	Vestibule, levator ani muscle, perineal body	VAS	4–24 months	After BoNT-A injections, it showed disappear of pain in all patients. Two patients needed only one injection; the other five patients received a 2nd injections VAS score improved from 8.3 to 1.4, with no recurrence. Improvement of sexual activity without significant discomfort during or after sexual intercourse.
Hebedo [59]	Prospective study	79	Dilution: 100 U of BoNT-A diluted into 1 mL isotonic saline. Dose: 100 U of BoNT-A	Bilaterally (50 units each site) and levator ani pars pubo rectalis	NRS NIQL Cotton swab test Active vita sexualis	6 months	Dyspareunia: 7.81 to 5.82 (*p* < 0.01) NIQL: 7.88 to 6.19 (*p* < 0.01) Cotton swab test: 6.81 to 5.50 (*p* < 0.01) Active Vitae Sexualis: no significances (*p* = 0.25)
Pelletier [61]	Prospective study	19	Dilution: 50 U of BoNT-A diluted into 1 mL saline. Dose: 100 U of BoNT-A	Bilateral bulbospongiosus muscles	VAS FSFI DLQI	24 months	Cured: 37% Mean VAS: 8.69 to 3.07 (*p* < 0.0001) Mean DLQI: 19.2 to 9.05 (*p* = 0.0005) Mean FSFI: 6.01 to 22.68 (*p* < 0.0001) Able to have sexual intercourse: 95%
Diomande [60]	Randomized, double-blind, placebo-controlled study	32	• Dilution: 50 U (arm A) or 100 U (arm B) of BoNT-A diluted in 1 mL saline. • Dose: • Baseline: Arm A: 50 U Arm B: 100 U Arm C: saline • 3 month-visit: Arm A & B & C: 100 U (for symptomatic patients) • 6 month-visit: Arm A & B: - Arm C: 100 U (for symptomatic patients)	Subcutaneous layers of the dorsal vestibulum (each side 0.5 mL)	Cotton swab-provoked VASVon Frey filamentsMarinoff dyspareunia scale	6–9 months	Improvement of cotton swab provoked VAS score: no significant difference between 3 groups and intragroup at 3 months. Improvement of von Frey filaments: significantly reduced pain level in all treatment groups including placebo arm after 3 months. It showed significant improvements of Marinoff dyspareunia scale between baseline and 3 months in arm A. Success rate (≥ 2 VAS point improvement): 58%

NRS: Numerical rating scale. NIQL: Negative interference in quality of life. FSFI: Female Sexual Function Index. DLQI: Dermatology Life Quality Index.

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
