# Peer review of "Can Botulinum Toxin A Play a Role in Treatment of Chronic Pelvic Pain Syndrome in Female Patients?—Clinical and Animal Evidence"

_toxins, 2020, doi:10.3390/toxins12020110_

Round 1

Reviewer 1 Report

The manuscript deals with chronic pelvic pain syndrome in women, potential mechanisms of botulinum toxin type A action and its efficacy.

In the manuscript introduction, CPP is described, however the reader is left without a clear definition of what does CPPS in general, or only in women stand for. Is the correct term a syndrome (as stated in the text) or syndroms? Does it encompass all possible syndromes in the pelvis area with a special referral to female population, or such CPPS which could occur only in women e.g. gynaecological pain etc?

Classification of the chronic pelvic pain is cited as a link to a shortened pocket version of European Association of Urology guidelines. Please cite original guidelines, and the guidelines from IASP from which the EAU guidelines are derived from. Can you please specify what does CPPS in women specifically mean? Is is referred to only gynecological or sexological types of CPPS etc.?

The potential mechanisms of BoNT-A action are stated together as valid for every type of CPPS. Maybe it would be better to separate a defined CPPS and link it to the most probable mechanism of action, since e.g. in bladder pain syndrome there is probably less involvement of BoNT action on skeletal muscle relaxation and, in contrary, there is probably less involvement of neuropeptide release inhibition in the treatment of pelvic floor spasm. Table 1 is titled as “Study of BoNT-A treatment for CPPS in the animal”. However, here the BoNT-A efficacy is reported only in chemically-evoked bladder pain, thus this title is misleading. Is it correct to generalize these data to all CPPS? Or the data is relevant only to IC/BPS? Maybe it would be better to bring together the data of a specific pain syndrome such as IC/BPS and connect it directly to the animal data which mainly concern bladder pain.

Minor:

Title: paly - play

Row 78-9"BoNT-A... first isolated in 1928. and its use could be traced back to the 18th century [12]" - this is illogical. Symptoms of botulism were first identified and described correctly at the beginning of 19th century, but its off-label use started in the 70ies.

Row 110 What does it mean „The effect of reversible paralysis can decrease noxious stimuli after BoNT-A injection.“?

Reviewer 2 Report

Modest English changes, perhaps better to be edited by a native English speaker.

Discuss if there are any clinical differences among the different subtypes of Botox-A products and whether there have been any studies with Type B

Round 2

Reviewer 1 Report

Minor comments:

In the title, a  correction "paly" to "play" was suggested, however this has not been corrected or rebutted.

(rows 117-120) After formation of antibodies which against BoNT-A, the duration of BoNT-A effect, and the therapeutic extent of the maximal treatment effect are usually reduced after few BoNT-A  applications (partial therapy failure) before complete therapy failure occurs. Dressler et al. reported the dose-effect relationship of BoNT-A treatment [31].

please correct the paragraph, since it is unclear or some words are missing. Citing of the the reference no 31 in the following sentence suggests that it is related to treatment failure. Maybe it can be placed  elsewhere in the text -  however,  the graphs provided in the reference (31) do not confirm correlation of the effect magnitude (% reduction of sternocleidomastoid muscle EMG) with the increasing doses.  
